# DEG-by-Index Ratio Transformation Normalization with Blood RNA-Seq Enhances Early and Consistent Detection of Mouse Tumorigenesis

**DOI:** 10.3390/biology14111577

**Published:** 2025-11-11

**Authors:** Sang Woon Shin, Ji Ae Kim, Jong-Hoon Kim, Jun Hyoung Jeon, Kunhyang Park, Dae-Soo Kim, Jong Soon Kang, Myeong Youl Lee, Doo-Sang Park, SooJin Lee, Hyun-Woo Oh

**Affiliations:** 1Independent Researcher, 1045 Whirlaway Lane, Cherry Hill, NJ 08002, USA; 2Core Research Facility & Analysis Center, Korea Research Institute of Bioscience and Biotechnology, Daejeon 34141, Republic of Korea; jiaekim@kribb.re.kr (J.A.K.); wjs258@korea.kr (J.H.J.); kunhyang@kribb.re.kr (K.P.); 3Department of Microbiology and Molecular Biology, Chungnam National University, Daejeon 34134, Republic of Korea; leesoojin@cnu.ac.kr; 4Microbiome Convergence Research Center, Korea Research Institute of Bioscience and Biotechnology, Daejeon 34141, Republic of Korea; kjh1018@pknu.ac.kr; 5Department of Biotechnology, College of Fisheries Science, Pukyong National University, Busan 48513, Republic of Korea; 6Department of Upland Crop Science, National Institute of Crop and Food Science, Rural Development Administration (RDA), Miryang 50424, Republic of Korea; 7Department of Digital Bio Technology Innovation, Korea Research Institute of Bioscience and Biotechnology, Daejeon 34141, Republic of Korea; kds2465@kribb.re.kr; 8Laboratory Animal Resource and Research Center, Korea Research Institute of Bioscience and Biotechnology, 30 Yeongudanjiro, Cheongju 28116, Republic of Korea; kanjon@kribb.re.kr (J.S.K.); myong@kribb.re.kr (M.Y.L.); 9Biological Resource Center, Korea Research Institute of Bioscience and Biotechnology, Jeongeup 56212, Republic of Korea; dspark@kribb.re.kr

**Keywords:** RNA-Seq, normalization, differentially expressed genes, tumorigenesis, DEG-by-index Ratio Transformation

## Abstract

Identifying early signals of tumors from blood using RNA-Seq is challenging because differences between samples can hide important changes in gene activity. In this study, we applied a new method called DiRT to analyze mouse blood RNA-Seq data. DiRT was able to clearly separate tumor samples from healthy ones at the earliest stages and consistently track changes as the disease progressed. In contrast, standard methods often detected differences only later or inconsistently. These results show that DiRT improves the sensitivity and reliability of blood RNA-Seq analysis, making it easier to detect tumor-related signals early in the disease.

## 1. Introduction

Recent advances in next-generation sequencing technologies have made RNA sequencing (RNA-Seq) a powerful tool for investigating transcriptomic differences between normal and diseased tissues [1]. Unlike reverse transcription polymerase chain reaction (RT-qPCR), RNA-Seq provides comprehensive expression profiles across the entire transcriptome, including the identification of differentially expressed genes (DEGs). In tumor-induced mouse (TIM) models—particularly in blood-based assays—transcript levels of disease-associated genes vary considerably across mice and time points, reflecting differences in tumor kinetics and tumor microenvironment composition [2,3,4]. Conventional normalization approaches aim to correct for global differences in sequencing depth and library composition, but often fail to account for gene-specific variability. This limitation reduces both the reproducibility and sensitivity of DEG detection, highlighting the need for improved normalization strategies that yield consistent and biologically meaningful results.

Normalization methods are essential for addressing both technical and biological variability inherent in RNA-Seq data [5]. Common strategies such as counts per million (CPM), transcripts per million (TPM), and fragments per kilobase per million (FPKM) adjust for the library size, sequencing depth, and gene length. More advanced statistical frameworks, including DESeq2 [6], edgeR [7], and limma/voom [8,9], further account for complex experimental designs. For instance, DESeq2 applies relative log expression (RLE) normalization, which compares each sample to a pseudo-reference generated from the median gene ratios, whereas edgeR uses the trimmed mean of M-values (TMM) to scale libraries by estimating and trimming extreme expression values. Although these global scaling approaches are widely used, they assume that most genes are not differentially expressed across samples—an assumption that may not hold true in heterogeneous or strongly perturbed biological systems such as tumorigenesis.

Recently, we developed an alternative RNA-Seq normalization method, the DEG-by-index Ratio Transformation (DiRT), which identified reproducible and biologically meaningful DEGs that were not detected using conventional global normalization approaches in *Drosophila* [10]. DiRT is conceptually related to compositional data analysis methods such as the additive log ratio (ALR) transformation [11]; however, a key distinction is that ALR applies a single reference gene or global denominator to all genes, whereas DiRT performs normalization locally by forming pairwise ratios between each DEG and a condition-insensitive index gene. The index gene is selected because it exhibits similar expression dynamics to the DEG under control conditions but remains unaffected by the experimental perturbation. This design allows DiRT to preserve the true biological signal of a DEG while minimizing unwanted variation from other sources.

Unlike global normalization methods that impose a uniform scaling factor across the transcriptome, DiRT performs pairwise normalization at the gene level. This localized transformation reduces dependence on global distributional assumptions and enhances sensitivity to modest yet biologically meaningful expression differences. By first statistically excluding index genes that show significant responses to the experimental condition and then systematically evaluating all possible DEG–index combinations, DiRT identifies reproducible differential patterns that may be diluted or lost when using conventional approaches.

In this study, we applied DiRT normalization to blood RNA-Seq datasets from the TIM model. By converting the read count (RC) of each DEG into a ratio with its designated index gene (RC_DEG/RC_index), DiRT reduced variability in blood RNA-Seq data and enabled the detection of early tumor-associated transcriptional changes as early as three days after tumor induction. This approach consistently identified marker signals across all stages of tumor progression, demonstrating that DiRT improves both the sensitivity and reproducibility of differential expression detection in complex, time-dependent biological systems.

## 2. Materials and Methods

### 2.1. Cell Culture and Animal Models

The murine colon adenocarcinoma cell line MC-38 (Kerafast, Cat. No. ENH204-FP; Boston, MA, USA) was cultured in Dulbecco’s modified Eagle’s medium (DMEM; Thermo Fisher Scientific, Waltham, MA, USA) supplemented with 10% fetal bovine serum (FBS; Thermo Fisher Scientific), 0.1 mM non-essential amino acids (Thermo Fisher Scientific), 100 U/mL penicillin (Roche, Basel, Switzerland), and 100 µg/mL streptomycin (Roche).

Six-week-old female C57BL/6N mice were purchased from Koatech (Pyeongtaek, Gyeonggi, Republic of Korea) and housed under specific pathogen-free conditions with a 12 h light/dark cycle at 21 ± 2 °C. Mice were acclimated for 1 week prior to experiments. All animal procedures were approved by the Institutional Animal Care and Use Committee (IACUC) of the Korea Research Institute of Bioscience and Biotechnology (Approval No. KRIBB-AEC-23137).

### 2.2. Tumor Induction and Sample Collection

MC-38 cells were harvested and resuspended in serum-free medium. Each mouse received a subcutaneous injection of 200 µL of either serum-free medium (control) or 200 µL containing 1 × 10^6^ MC-38 cells. At 3, 7, and 10 days post-injection, 0.5 mL of blood was collected via retro-orbital puncture into heparinized tubes, immediately mixed with 1.3 mL RNAlater solution (Thermo Fisher Scientific, San Jose, CA, USA), and stored at −70 °C. RNA was extracted using the Mouse RiboPure™ Blood RNA Isolation Kit (Thermo Fisher Scientific, San Jose, CA, USA) following the manufacturer’s instructions.

### 2.3. RNA Quality Control and Sequencing

Total RNA isolated from mouse blood was assessed for quality using the Agilent 2100 Bioanalyzer (Agilent Technologies, Santa Clara, CA, USA). Only samples with an RNA integrity number (RIN) ≥ 7.5 were used for library preparation. RNA-Seq libraries were constructed using the Illumina TruSeq Library Preparation Kit (Illumina, San Diego, CA, USA) according to the manufacturer’s protocol. Prepared libraries were diluted to 2 nM using Resuspension Buffer with Tween 20 prior to sequencing.

Sequencing was performed on an Illumina NextSeq 1000 platform (Illumina) with a read length of 1 × 100 bp, following the standard Illumina RNA-Seq workflow. Raw reads were evaluated for quality using NGSQC Toolkit v2.3.3. Adapter sequences were removed with Cutadapt v3.7 using default parameters, and low-quality bases were trimmed with Sickle v1.33 at a Phred quality threshold of 20. Reads containing ambiguous bases (N) or those shorter than 50 bp after trimming were discarded.

### 2.4. Data Processing and Read Mapping

A total of 111 mouse blood RNA-Seq datasets were analyzed using a uniform workflow implemented on the Galaxy server (Galaxy v19.09; usegalaxy.org) [12]. Single-end reads were preprocessed using fastp (Galaxy Version 1.0.1 + galaxy1) [13] to trim adapters and remove low-quality bases. Cleaned reads were aligned to the mouse reference genome GRCm39 [14] with HISAT2 (Galaxy Version 2.2.1 + galaxy1) [15]. The resulting BAM files were quantified with featureCounts (Galaxy Version 2.1.1 + galaxy0) [16] using the NCBI reference gene annotation (GCF_000001635.27).

### 2.5. DESeq2/edgeR Normalization Analyses

After computing counts per million (CPM) across the 111 datasets, the 10,000 most abundantly expressed genes (CPM > 2.27) were selected from the TIM dataset for subsequent normalization analyses using RLE (DESeq2) and TMM (edgeR). This filtering minimized stochastic variation from low-abundance transcripts.

Differential expression analyses were performed in R (v4.3.1) with DESeq2 (v1.40.2) and edgeR (v3.42.4). For DESeq2, normalization was performed using the RLE method, while edgeR used the TMM method. For both approaches, differential expression was assessed using a two-sample, two-tailed *t*-test assuming equal variance, identical to the statistical procedure applied in the DiRT analysis. This consistent framework allowed for direct comparison across methods without bias from model-specific implementations. The gene-level *p*-values were adjusted using the Benjamini–Hochberg procedure, and the ranking of the most significant genes was highly concordant with the results from each package’s default model.

### 2.6. Computation of DiRT Values

For each gene pair consisting of a target gene (potential DEG) and a candidate index gene, DiRT calculates the ratio of their read counts (RC_target/RC_index) within each sample. Candidate index genes are chosen because their expression profiles closely resemble those of the target gene under control conditions yet remain largely unaffected by the experimental perturbation. This pairwise ratio transformation expresses each gene’s abundance relative to its contextually stable reference, thereby reducing sample-dependent variability while preserving biologically relevant differences.

To ensure reliable reference selection, DiRT quantifies expression similarity using the normalized standard deviation (NSD) across all control samples. Candidate index genes with low NSD values and no significant response to the condition (adjusted *p* > 0.05 via two-tailed *t*-test with Benjamini–Hochberg correction) were considered suitable. For each target gene, all valid DEG–index pairings were evaluated, and the smallest adjusted *p*-value among these pairings defined the DiRT significance for that gene.

Unlike global scaling methods such as RLE, TMM, or ALR, which assume uniform expression stability, DiRT performs normalization on a pairwise basis anchored to empirically determined, condition-insensitive reference genes. This localized framework reduces sensitivity to outliers, batch effects, and compositional bias, thereby improving reproducibility across samples and time points. Additional normalization and compositional strategies for comparison are described elsewhere [17,18].

### 2.7. DiRT Analyses for the TIM Dataset

DiRT analyses were performed as previously described [10], with modifications to adapt the method to the mouse TIM dataset. For each of the 10,000 abundantly expressed genes (CPM > 2.27), the ratio of its RC to the RC of every other gene was calculated. From these pairwise comparisons, the ten gene pairs with the lowest NSD values within the control group (*n* = 54) were selected as candidate index genes.

A DiRT candidate database comprising 100,000 target–index gene pairs was generated using a custom Python script (version 3.0) available at https://github.com/shinwoongg/DiRT-normalization (accessed on 9 November 2025). For each target–index pair, statistical significance between control and TIM groups was evaluated using a two-tailed, equal-variance *t*-test applied to DiRT-normalized values (RC_target/RC_index). Multiple testing correction was conducted using the Benjamini–Hochberg method across all 100,000 comparisons.

Pairs involving index genes that showed significant responses to tumor induction (adjusted RLE-based *p* < 0.05) were excluded, resulting in a final set of 33,272 target–index pairs. Unlike global normalization methods that apply a uniform scaling factor across the transcriptome, DiRT normalizes expression at the gene pair level. This localized transformation reduces dependence on global distributional assumptions and increases sensitivity to subtle but biologically meaningful expression changes. By systematically filtering condition-responsive index genes and evaluating all remaining DEG–index combinations, DiRT identifies reproducible differential patterns that may be diluted or overlooked when using conventional normalization approaches.

To assess DiRT performance relative to established normalization methods, the same TIM RNA-Seq datasets were analyzed using RLE/DESeq2 [6] and TMM/edgeR [7] under identical statistical conditions. Comparisons were based on (i) the number and overlap of DEGs detected at equivalent false discovery thresholds, (ii) the reproducibility of DEG detection across time points (days 3, 7, and 10 after tumor induction), and (iii) biological interpretability evaluated through KEGG pathway enrichment. The results demonstrated that DiRT provided improved sensitivity in early-stage detection and greater reproducibility of tumor-related transcriptional signals in blood RNA-Seq data compared with conventional global normalization methods.

### 2.8. Heatmap Generation

Heatmaps were generated to visualize the expression profiles of the 100 genes or 100 gene–index pairs (DiRT) with the lowest adjusted *p*-values across four normalization methods: DiRT, CPM, RLE/DESeq2, and TMM/edgeR. Expression values were normalized relative to the average expression across all RNA-Seq samples. Heatmaps were created in Python using the libraries matplotlib, seaborn, and pandas. In the heatmaps, red indicates expression values greater than 1 (above average), while blue represents values lower than 1 (below average). Thus, induction events appeared as blue/red patterns in controls/TIMs, with the inverse pattern observed vice versa.

### 2.9. Null Hypothesis Testing

An artificial TIM dataset was generated for the null set by combining RNA-Seq RC data from the 30 control samples (10 samples from 3 time points) and 27 TIM samples (9 samples from 3 time points) to create artificial control data. The remaining 27 control samples (9 samples from 3 time points) and 27 TIM samples (9 samples from 3 time points) were designated as artificial disease data.

### 2.10. Statistical Analysis

The *p*-values were calculated using a two-tailed, two-sample equal-variance *t*-test to compare control and TIM samples. Multiple-testing correction was performed using the Benjamini–Hochberg method. The ranks applied for adjustment were 10,000 for RLE-normalized genes, 10,000 for TMM-normalized genes, and 100,000 for the DiRTs (original or null).

## 3. Results

### 3.1. DiRT Normalization of TIM RNA-Seq Data

Figure 1 presents an overview of the experimental design and DiRT analysis pipeline. The left panel depicts the control and tumor-induced (TIM) mouse groups, with blood samples collected at days 3, 7, and 10 post MC-38 injection (a total of 111 RNA-Seq samples). The middle panel illustrates the data-processing workflow—from raw reads processed through fastp, HISAT2, and featureCounts—to the generation of a gene-count matrix and selection of 10,000 abundantly expressed genes for DiRT analysis. The right panel summarizes the DiRT normalization procedure, including computation of 100,000 target–candidate index gene pairs, NSD-based index selection, and identification of valid pairs with low adjusted *p*-values.

Orthotopic subcutaneous tumors [19] were induced in 20 mice (Mus musculus) by injecting MC-38 colon adenocarcinoma cells, while 20 control mice received serum-free medium. The MC-38 adenocarcinoma colorectal cell line is a commonly employed tumor model for preclinical studies of neoantigens and immunotherapeutic approaches [20]. All mice injected with MC-38 cells developed visible tumors at the injection sites within 3 weeks, whereas no visible tumors were observed in control mice injected with serum-free medium. Whole blood was collected on days 3, 7, and 10 post-injection, yielding high-quality RNA suitable for RNA-Seq from 19 control mice and 18 TIM. In total, 111 transcriptomes (57 control and 54 TIM samples) were sequenced, generating 22–37 million reads per sample mapped to annotated genes in the GRCm39. NCBI SRA accession numbers for all datasets are provided in Appendix A.

DiRT normalization was applied to the TIM RNA-Seq dataset. First, 10,000 abundantly expressed genes with an average CPM exceeding 2.27 across 111 datasets were selected. CPM normalization, performed after RC quantification of annotated GRCm39 genes, scaled RCs by the total library size to generate “reads per million.” All further analyses were restricted to these 10,000 genes to identify suitable combinations of DEGs and index genes for DiRT normalization while minimizing false positives arising from unstable RCs of low-abundance transcripts.

Next, 10 candidate index genes were assigned to each target gene by selecting those with the lowest NSD of expression ratios across 57 control samples. Specifically, the RC of each gene was divided by that of the other 9999 genes, and the NSD of each ratio (RC_target gene_/RC_other genes_) was computed. The ten lowest NSD ratios per gene were retained, generating a database of 100,000 DiRT candidates via a Python script. To assess differential expression, *t*-tests were performed to evaluate significant differences in RC_target gene_/RC_other gene_ between 57 control and 54 TIM samples, and the resulting *p*-values were adjusted using the Benjamini–Hochberg method with 100,000 ranks.

Finally, only target gene–index gene pairs were retained to construct the TIM DiRT database. Differential expression was evaluated using a *t*-test on RLE-normalized values between 57 control samples (19 × 3) and 54 TIM samples (18 × 3). The resulting *p*-values for 10,000 genes were adjusted using the Benjamini–Hochberg method with 10,000 ranks, representing 20% of the mouse transcriptome (10,000/49,997). From the DiRT candidate database, we retained only pairs in which the index genes had adjusted *p*-values > 0.05, ensuring that the index genes were unaffected by tumorigenesis. This filtering step excluded the majority of initial DiRTs, yielding a final set of 33,272 DiRTs composed exclusively of target gene–index gene pairs (Appendix A).

### 3.2. Heatmap Shows the Advantages of DiRT Normalization

DESeq2 and edgeR are widely used tools for differential expression analysis, alongside employing internal normalization methods: RLE in DESeq2 and TMM in edgeR. The application of these normalization strategies improved discrimination between control and TIM samples compared with CPM normalization. Consequently, the adjusted *p*-values for the identified DEGs were substantially lower with RLE and TMM than with CPM (CPM: 4.61 × 10^−11^ to 1.99 × 10^−8^; RLE: 1.39 × 10^−23^ to 2.62 × 10^−13^; TMM: 2.56 × 10^−22^ to 4.65 × 10^−13^, based on the 100 DEGs with the lowest adjusted *p*-values). Notably, DiRT normalization demonstrated clear advantages over both RLE and TMM approaches (Figure 2).

Heatmaps of the top 100 features with the lowest adjusted *p*-values revealed differences between normalization strategies. DiRT normalization (Figure 2a) produced consistent separation between control and TIM samples across all three time points—3, 7, and 10 days post-tumorigenesis. In contrast, CPM measures (Figure 2b) provided almost no discrimination at 3 days, with most signals appearing repressed. The conventional approaches, RLE/DESeq2 (Figure 2c) and TMM/edgeR (Figure 2d), achieved clear separation between groups at 7 and 10 days, but only marginal discrimination at 3 days. Together, these findings suggest that DiRT provides an advantage in detecting early and consistent transcriptomic changes compared with conventional normalization methods.

### 3.3. The Expression Profiles of Gimap 6 and Mylip After RLE/TMM/CPM Normalization

Figure 3 shows the expression patterns of two DEGs with the lowest adjusted *p*-values, Gimap6 (GTPases of the immunity-associated protein 6) and Mylip (myosin-regulated light chain interacting protein), under RLE or TMM normalization (Appendix A). Gimap6 is essential for peripheral T-cell maintenance and efficient autophagy; mice with lymphocyte-specific Gimap6 deletion exhibit T-cell loss and autophagic defects, suggesting that reduced Gimap6 expression could impair anti-tumor immune surveillance and thereby promote tumor initiation/progression in mice [21,22]. In contrast, Mylip encodes a RING E3 ubiquitin ligase with tumor-modulating activity: in lung cancer models, Mylip expression is decreased in tumors, and enforced expression suppresses proliferation, invasion, and xenograft growth in nude mice, consistent with a tumor suppressive role [23].

Both Gimap6 and Mylip showed clear expression differences between controls and TIM samples at later stages of mouse tumorigenesis under RLE and TMM normalization. For *Gimap6*, RLE normalization yielded *p*-values of 1.10 × 10^−14^ (7 days) and 3.86 × 10^−18^ (10 days), while TMM normalization produced *p*-values of 3.09 × 10^−13^ (7 days) and 6.86 × 10^−18^ (10 days). For *Mylip*, the corresponding values were 4.56 × 10^−13^ (7 days) and 1.49 × 10^−16^ (10 days) under RLE normalization, and 6.76 × 10^−12^ (7 days) and 1.27 × 10^−16^ (10 days) under TMM normalization.

At the earlier stage (3 days post-tumorigenesis), both RLE- and TMM-normalized measures showed smaller differences between control and TIM samples than at later stages, although the *p*-values (10^−5^ range) remained statistically significant. For *Gimap6*, RLE normalization yielded a *p*-value of 2.71 × 10^−5^, while TMM normalization produced 3.78 × 10^−5^. For *Mylip*, the corresponding values were 1.09 × 10^−5^ under RLE normalization and 1.01 × 10^−5^ under TMM normalization.

Variation associated with sampling time points was also observed in the expression of both *Gimap6* and *Mylip*. In control samples, expression levels of these genes varied markedly across time points, showing greater variation than the differences observed between control and TIM samples at 3 days post-tumorigenesis (Figure 3).

### 3.4. DiRT Normalization for Better Discrimination at Early Stages of Tumorigenesis

While RLE and TMM normalization revealed clear expression differences for *Gimap6* and *Mylip* at later stages (7 and 10 days), their ability to distinguish control from TIM samples was limited at the early stage (3 days post-tumorigenesis). In contrast, DiRT normalization provided clear separation even at this earliest time point, with heatmaps showing distinct clustering of control and TIM samples. Notably, DiRT maintained consistent group separation across all time points (3, 7, and 10 days), whereas conventional methods achieved strong separation only at the later stages.

Figure 4 highlights two DiRT-normalized DEG–index gene pairs with the lowest adjusted *p*-values. *Hbq1b/Gm6055* (Figure 4a) and *Plekhg3/Sh2d3c* (Figure 4b) were identified as the top-ranked pairs (Appendix A). Using RLE normalization, the DEGs *Hbq1b* and *Plekhg3* showed adjusted *p*-values of 0.00020 and 2.96 × 10^−12^, respectively, whereas their corresponding index genes, *Gm6055* and *Sh2d3c*, had adjusted *p*-values of 0.52 and 0.31. These values satisfy the criterion for index genes (adjusted *p*-value > 0.05), suggesting that they were not directly affected by tumor induction.

DiRT normalization (Figure 4a,b) provided clearer discrimination between control and TIM samples at the early stage (3 days post-tumorigenesis) compared with conventional normalization approaches such as RLE/DESeq2 and TMM/edgeR (Figure 3). At 3 days, DiRT yielded *p*-values of 1.18 × 10^−12^ for *Hbq1b/Gm6055* and 4.64 × 10^−7^ for *Plekhg3/Sh2d3c*. In contrast, the two DEGs, *Gimap6* and *Mylip*, previously identified as having the lowest adjusted *p*-values under RLE or TMM normalization (Figure 3), produced *p*-values of 1.01 × 10^−5^ and 3.78 × 10^−5^, respectively, at the same time point.

In addition, we examined two DiRT-normalized DEG–index gene pairs, *Gimap6/Rsbn1l* (Figure 4c) and *Mylip/Cyld* (Figure 4d), corresponding to the RLE/TMM-normalized DEGs *Gimap6* and *Mylip*. During index gene selection, DiRT normalization initially failed to identify suitable index genes with adjusted *p*-values > 0.05 among the first 10 candidates; therefore, no DiRTs with *Gimap6* or *Mylip* as DEGs were included in the final set of 33,272 DiRTs (Appendix A). Extending the manual DEG–index pairing to 100 candidate index genes identified valid DiRTs with lower adjusted *p*-values. Although DiRT normalization only modestly improved early-stage (3 days) discrimination compared with RLE-normalized measures (*p*-values decreased from 2.71 × 10^−5^ for Gimap6 to 8.82 × 10^−6^ for *Gimap6/Rsbn1l*, and from 1.09 × 10^−5^ for Mylip to 2.72 × 10^−6^ for *Mylip/Cyld*), it produced more consistent normalized measures across control samples (Figure 5). Similar improvements in consistency were observed for *Hbq1b/Gm6055* and *Plekhg3/Sh2d3c* (Figure 5). Notably, whereas control measures exhibited time point-related variation under RLE normalization, DiRT normalization reduced this variability and yielded more stable values. Overall, DiRT normalization not only enhanced early-stage discrimination during mouse tumorigenesis but also more effectively normalized time point-dependent variation between control samples compared with RLE/DESeq2.

DiRT normalization methods could be adapted for disease diagnostics using blood RNA-Seq, given their unique characteristics. In this study, the application of DiRT to mouse blood RNA-Seq data resulted in clearer discrimination between control and TIM samples. Moreover, DiRT normalization minimized time point–related variation effects among controls and reduced within-group variance, thereby enabling more reliable separation of TIM samples. These findings suggest that discrimination based on DiRT offers stronger power to distinguish between control and TIM groups. For the best performing DiRT pair, *Hbq1b/Gm6055*, the maximum control value was 0.9397 for the day 3 C16 sample, which exceeded values observed in 48 of 54 TIM samples (Appendix A). In contrast, the highest control value of the *Gimap6* measure after RLE normalization was 6122 for the day 3 C12 sample, which surpassed many TIM samples and thus demonstrated limited discriminatory capacity during mouse tumor development (Appendix A).

### 3.5. KEGG Pathway Analysis of DiRT-Derived DEGs

In a previous Drosophila study [10], DiRT normalization was applied to identify DEGs after methyl lucidone treatment and revealed a distinct set of DEGs with higher reproducibility and more reliable validation compared with RLE/TMM normalization (10). The DiRT-derived DEGs were enriched in KEGG pathways associated with methyl lucidone detoxification, suggesting that DiRT can uncover putative DEGs not detected using conventional normalization methods.

In the current mouse tumor progression dataset, DiRT-derived DEGs showed limited overlap with those identified using RLE/TMM normalization (Figure 6). From the 1000 DiRTs with the lowest adjusted *p*-values (Appendix A), duplicate target genes were removed to yield 654 unique DEGs. For comparison, the 1000 DEGs with the lowest FDR were compiled from RLE/TMM. Of the 654 DiRT-derived DEGs, 406 (62.1%) were not present in the RLE/TMM set, leaving 248 genes in common. In contrast, the RLE and TMM lists themselves were highly concordant, sharing 925 of their top 1000 DEGs (92.5% overlap, Figure 6).

KEGG analysis of the 406 DiRT-specific DEGs revealed enrichment of platelet activation signaling, including key genes such as *ITGA2B/αIIb*, *P2RY1/P2Y12*, *TBXAS1*, *RAP1A/B*, and *RGS18* (Table 1). These genes mediate ADP-driven platelet activation, integrin αIIbβ_3_ signaling, thromboxane A_2_ synthesis, and RAP1-dependent integrin activation. Platelet activation has been implicated in tumor immune evasion, vascular priming, and metastatic dissemination [24], and inhibition or genetic deletion of *P2Y12* reduces metastasis and improves survival in murine models [25]. The observed enrichment is consistent with evidence that platelets are rapidly recruited to disseminated tumor cells to form early metastatic niches via secretion of CXCL5/7 [26], supporting the ability of DiRT normalization to detect platelet activation signatures associated with early tumorigenesis. In contrast, KEGG analysis of the 209 DEGs consistently identified across all three normalization methods (DiRT, RLE, and TMM) did not reveal any significantly enriched pathways (adjusted *p* < 0.05).

In comparison, KEGG analysis of the 716 DEGs obtained after RLE/TMM normalization predominantly revealed pathways already well established in immune regulation and tumorigenesis (Table 1). Many enriched terms corresponded to innate and adaptive immune signaling, including NOD-like receptor signaling, T-cell receptor signaling, Th17 cell differentiation, hematopoietic cell lineage, and NK cell-mediated cytotoxicity [27,28]. Numerous infection-related pathways were also overrepresented, such as viral (Influenza A, Epstein–Barr virus, herpes simplex virus 1, hepatitis C, measles, HIV-1 life cycle, Kaposi sarcoma-associated herpesvirus, COVID-19) and bacterial or parasitic pathways (Yersinia infection, tuberculosis, toxoplasmosis, leishmaniasis) [27,29]. In addition, several cancer- and inflammation-related pathways were detected, including PD-L1/PD-1 checkpoint signaling, NF-κB signaling, lipid and atherosclerosis, regulation of the actin cytoskeleton, FcγR-mediated phagocytosis, and sphingolipid metabolism/signaling [28,30,31]. Collectively, these findings suggest that RLE/TMM-derived DEGs primarily reflect broad immune activation and infection-related processes, which are already widely linked to tumor progression.

### 3.6. Null Hypothesis Testing for TIM DiRT

When calculating the NSD of expression ratios between DEGs and index genes, only control samples were used for index gene identification, with TIM samples excluded. Based on this design, we hypothesized that the markedly lower adjusted *p*-values observed for DiRT reflected effective normalization; however, such improvements could occur by chance. To evaluate this null hypothesis, an artificial RNA-Seq dataset was generated by combining 30 control samples (ten per time point) with 27 TIM samples (nine per time point) to create artificial controls, while the remaining 27 control and 27 TIM samples (nine per time point each) served as artificial TIM data (Appendix A). Analysis of the artificial dataset yielded no DiRTs with adjusted *p*-values < 0.1. In contrast, in the original dataset, the 1000 lowest-ranked DiRTs showed adjusted *p*-values ranging from 1.17 × 10^−25^ to 1.60 × 10^−9^, whereas in the null dataset, values were restricted to 0.11–0.28 (Figure 7). These results indicate that the low adjusted *p*-values achieved by DiRT normalization did not arise by chance, thereby rejecting the null hypothesis.

## 4. Discussion

KEGG analysis of the 209 DEGs identified across all three normalization methods (DiRT, RLE, and TMM) did not reveal any significantly enriched pathways. However, these genes are likely to represent true DEGs, as they were reproducibly detected regardless of normalization strategy. The absence of significant pathway enrichment reflects current limitations in understanding the molecular events underlying early tumorigenesis, suggesting that relevant processes remain unannotated or unrecognized in existing pathway databases.

We observed that DEG–DEG pairs with opposite regulatory directions occasionally generated DiRT values with low adjusted *p*-values. Such combinations likely represent artificial pairings that coincidentally satisfied the selection criteria. To minimize their occurrence, we applied a *t*-test *p*-value threshold (>0.05) to the index gene, using adjusted *p*-values calculated after RLE normalization. This phenomenon likely arises because dividing the expression of an upregulated gene by that of a downregulated gene (or vice versa) can artificially increase the expression contrast between groups, producing exceptionally small, adjusted *p*-values. In many practical applications, the inclusion of a small number of such pairs is unlikely to affect the main biological conclusions, as they do not dominate the ranking of top candidates. Notably, these cases were rare and were previously observed in the *Drosophila* DiRT dataset, where two of the top ten DiRTs involved DEG–DEG combinations. No comparable instances were found in the current mouse (TIM) dataset after the filtration of index genes, indicating that this artifact has minimal influence on the overall performance of DiRT. Nevertheless, awareness of their potential presence remains important for accurate interpretation of DiRT results.

An adjusted *p*-value cutoff was not imposed on target genes/DEGs, as no threshold could be reasonably justified in this context. Consequently, it cannot be concluded that all target genes identified by DiRT represent DEGs. Importantly, even a few outlier data points substantially affected *p*-value estimation in RLE analyses, suggesting that applying an arbitrary DEG filter could result in the loss of valuable information. Nonetheless, among the 100 target genes forming the 100 DiRTs with the lowest adjusted *p*-values, 98 exhibited adjusted *p*-values < 0.001 under RLE normalization (Appendix A), strongly supporting the interpretation that these DiRTs—with extremely low adjusted *p*-values ranging from 1.17 × 10^−25^ (1st) to 7.08 × 10^−15^ (100th)—represent true DEG–index gene pairs.

We also performed additional functional analyses using the Database for Annotation, Visualization, and Integrated Discovery (DAVID) [32], complementing the KEGG pathway analysis. The DAVID knowledgebase substantially enriches the biological context of a given gene by integrating diverse annotations, including gene/protein identifiers, functional domains, Gene Ontology terms, pathways, disease associations, general descriptions, protein–protein interactions, the literature links, and small-molecule interactions. The complete bioinformatic results generated by DAVID for the 406 DiRT-only DEGs, 209 DiRT/RLE/TMM shared DEGs, and 716 RLE/TMM-only DEGs are summarized in Appendix A.

Although the present study focused on blood RNA-Seq data, the effectiveness of DiRT is likely not limited to this tissue type. Blood consists of diverse and dynamic cell populations, leading to substantial variability in gene expression. The ability of DiRT to consistently normalize such complex datasets indicates its potential applicability to RNA-Seq analyses from other organs or experimental systems with diverse cellular compositions. In our previous work using whole-larva RNA-Seq data from Drosophila melanogaster, DiRT also maintained stable performance, supporting its general utility in complex biological contexts. Therefore, DiRT is expected to maintain its performance in RNA-Seq data derived from other tissues or disease models.

## 5. Conclusions

DiRT normalization of blood RNA-Seq in the TIM model yielded clearer separation of control and tumor samples than conventional methods, reduced time point-related variability, and enabled early detection of tumor-associated transcriptomic changes. These findings support the potential of DiRT as a sensitive approach for blood-based tumor diagnostics.

## Figures and Tables

**Figure 1 biology-14-01577-f001:**
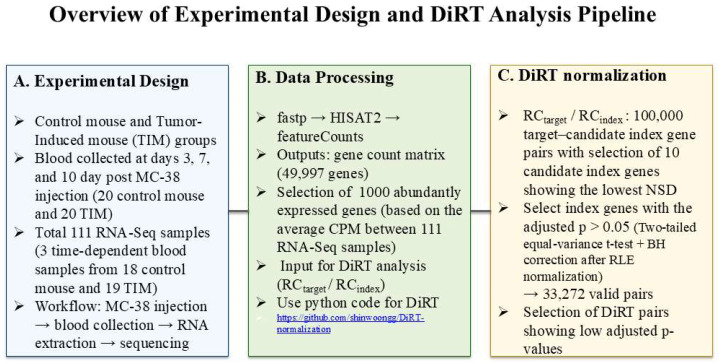
Overview of experimental design and DiRT Analysis pipeline.

**Figure 2 biology-14-01577-f002:**
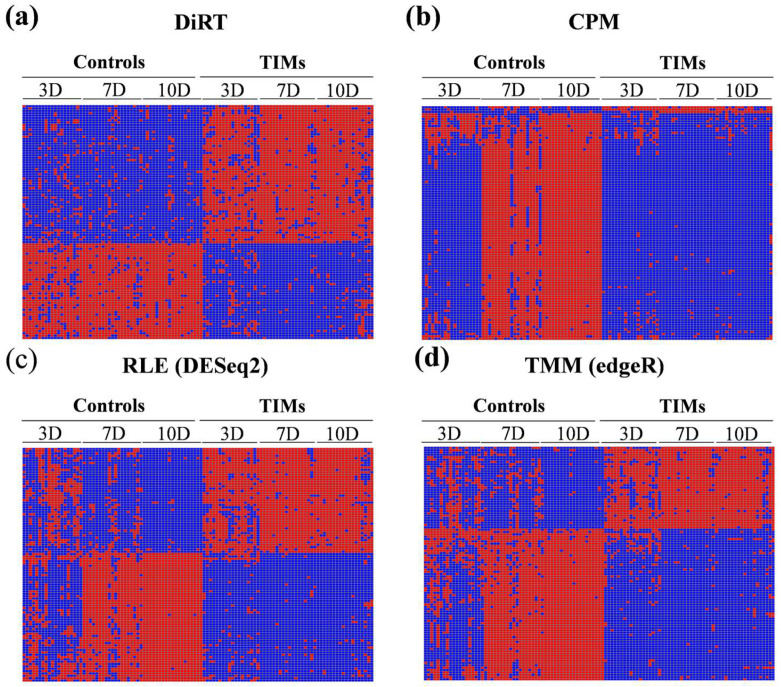
Heatmaps of the top 100 signals with the lowest adjusted *p*-values for each method: (**a**) DiRT-derived DEG–index ratios; (**b**) CPM; (**c**) RLE/DESeq2; (**d**) TMM/edgeR. Inducible signals (DEGs for CPM/RLE/TMM and DiRT-derived ratios for DiRT) are shown in blue for control samples and red for TIM samples, whereas repressive signals display the opposite pattern. DiRT consistently separated control and TIM samples across all three time points (3, 7, and 10 days post-tumorigenesis), whereas RLE/TMM achieved clear separation only at 7 and 10 days, with marginal discrimination at day 3.

**Figure 3 biology-14-01577-f003:**
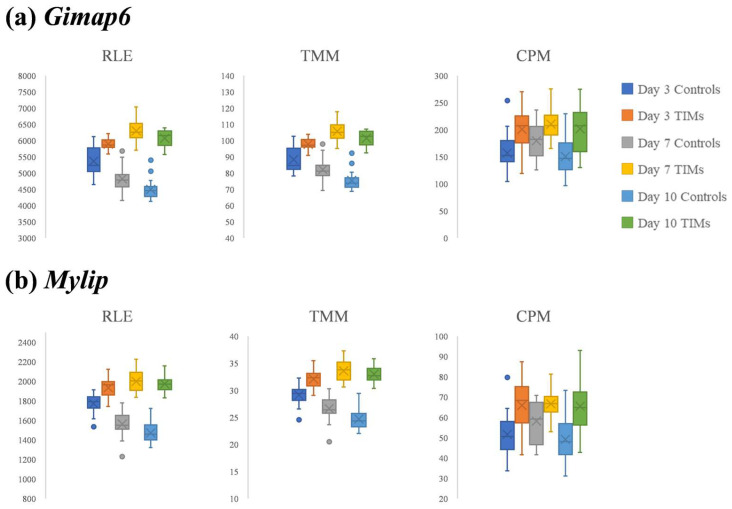
Boxplots of (**a**) *Gimap6* and (**b**) *Mylip* expression in control and TIM samples at 3, 7, and 10 days post-tumorigenesis using three normalization methods: RLE (left panels), TMM (middle panels), and CPM (right panels). Both genes exhibited clearer separation between control and TIM samples with RLE and TMM than with CPM, particularly at 7 and 10 days.

**Figure 4 biology-14-01577-f004:**
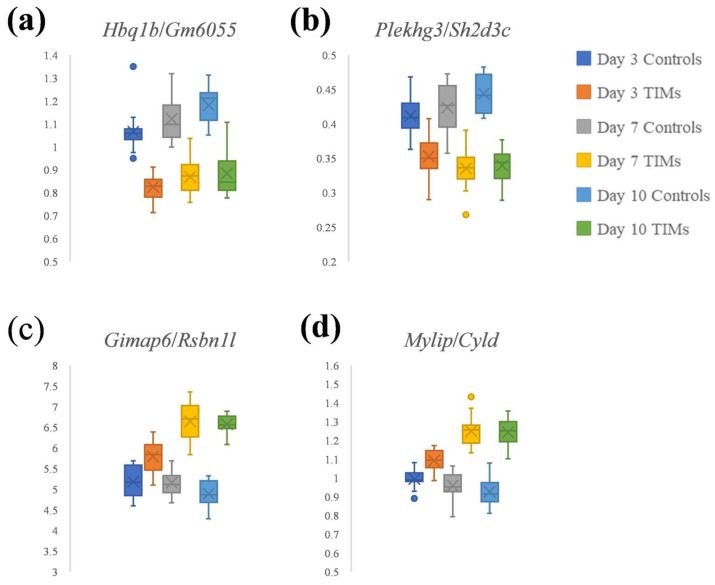
Boxplots of DiRT normalization: (**a**) *Hbq1b/Gm6055*; (**b**) *Plekhg3/Sh2d3c*; (**c**) *Gimap6/Rsbn1l*, and (**d**) *Mylip/Cyld*. *Hbq1b/Gm6055* and *Plekhg3/Sh2d3c* represent the top-ranked DEG–index gene pairs with the lowest adjusted *p*-values (Appendix A). *Gimap6/Rsbn1l* and *Mylip/Cyld* correspond to additional DiRT normalization of two RLE-normalized DEGs.

**Figure 5 biology-14-01577-f005:**
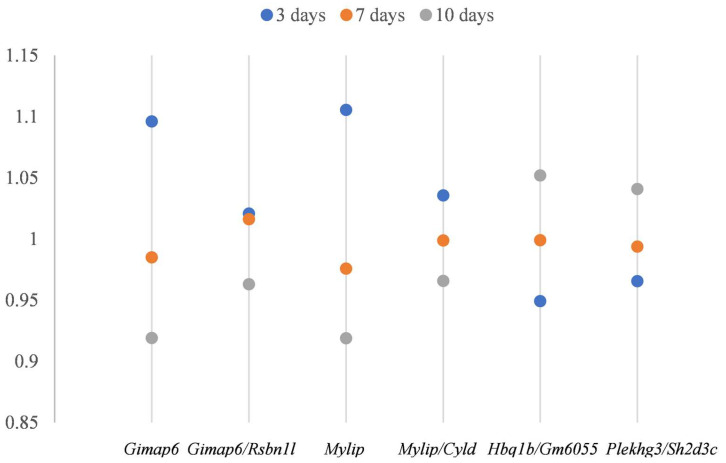
Normalized average expression in control samples across 3, 7, and 10 days: Variation across ages was evident in RLE-normalized DEGs (Gimap6 and Mylip), whereas DiRT-normalized pairs (*Gimap6/Rsbn1l*, *Mylip/Cyld*, *Hbq1b/Gm6055*, and *Plekhg3/Sh2d3c*) reduced this variation and provided more consistent control measures.

**Figure 6 biology-14-01577-f006:**
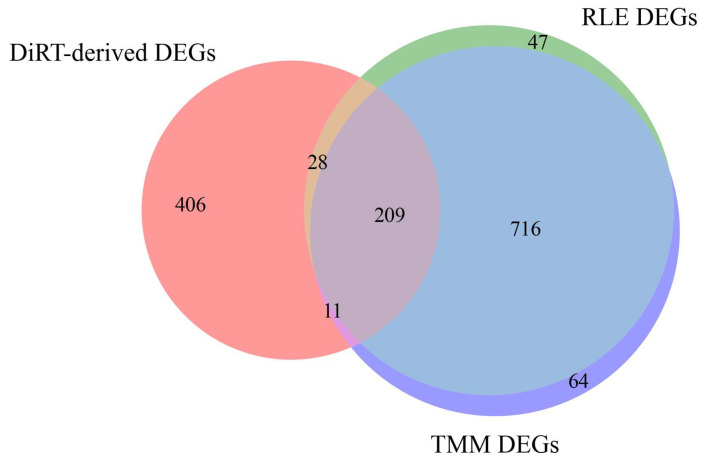
A Venn diagram comparing DEGs detected using DiRT (red), RLE (green), and TMM (violet) approaches: Among the 654 unique DiRT-derived DEGs, 406 were specific to DiRT, 28 overlapped with RLE only, 11 overlapped with TMM only, and 209 were shared across all three methods. RLE and TMM were largely concordant, sharing 925 of their top 1000 DEGs.

**Figure 7 biology-14-01577-f007:**
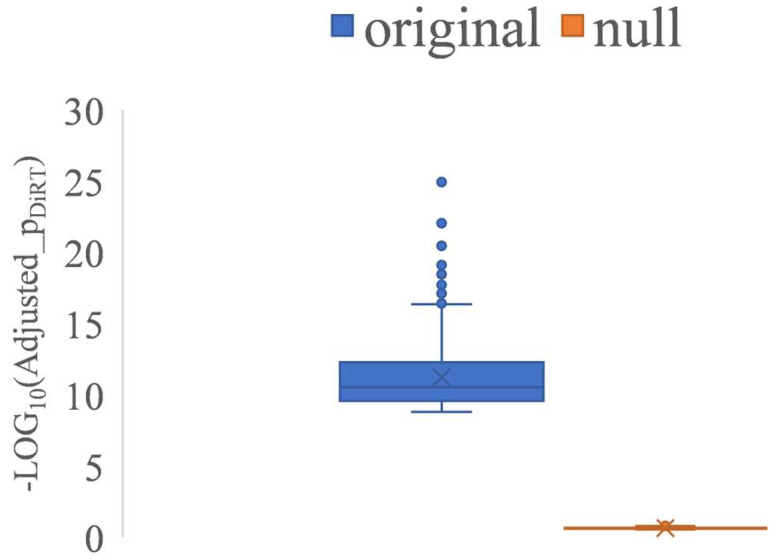
A comparison of adjusted *p*-values for the 1000 top-performing DiRTs between the original and null datasets. The *Y*-axis represents –log_10_ of the adjusted *p*-values.

**Table 1 biology-14-01577-t001:** **KEGG pathway enrichment of DEGs:** DiRT-specific DEGs were enriched in platelet activation. Shared DEGs showed no significant enrichment, whereas RLE/TMM-specific DEGs were enriched in broad immune and infection pathways.

KEGG Pathway	Adjusted *p*
**DiRT only: 406 genes**	
Platelet activation	0.000037
**DiRT/RLE/TMM: 209 genes**	
None	
**RLE/TMM only: 716 genes**	
NOD-like receptor signaling pathway	1.5 × 10^−7^
Primary immunodeficiency	0.000056
Influenza A	0.000056
Yersinia infection	0.000056
Tuberculosis	0.00021
PD-L1 expression and PD-1 checkpoint pathway in cancer	0.00021
Epstein–Barr virus infection	0.00021
Measles	0.0003
Th17 cell differentiation	0.0012
Hematopoietic cell lineage	0.0014
Osteoclast differentiation	0.0014
Herpes simplex virus 1 infection	0.0038
Hepatitis C	0.0038
T-cell receptor signaling pathway	0.0038
Lipid and atherosclerosis	0.0038
Fc gamma R-mediated phagocytosis	0.0038
Toxoplasmosis	0.0039
Leishmaniasis	0.0046
Regulation of actin cytoskeleton	0.0081
Phagosome	0.0081
Coronavirus disease—COVID-19	0.012
Sphingolipid metabolism	0.013
Human immunodeficiency virus 1 infection	0.025
NF-kappa B signaling pathway	0.028
Bacterial invasion of epithelial cells	0.028
Viral life cycle—HIV-1	0.028
Natural killer cell-mediated cytotoxicity	0.028
Sphingolipid signaling pathway	0.03
Kaposi sarcoma-associated herpesvirus infection	0.047

## Data Availability

The RNA-Seq datasets of the 111 TIM samples have been deposited in the NCBI SRA database. The accession numbers are indicated in Appendix A. The Python script used to generate DiRT candidate databases is available in the GitHub repository [https://github.com/shinwoongg/DiRT-normalization (accessed on 9 November 2025)].

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
