# Peer review of "DEG-by-Index Ratio Transformation Normalization with Blood RNA-Seq Enhances Early and Consistent Detection of Mouse Tumorigenesis"

_biology, 2025, doi:10.3390/biology14111577_

Round 1
Reviewer 1 Report
Comments and Suggestions for Authors
- Lines 240, 275: On day 3, the p-values detected for genes like Gimap6 and Mylip under RLE and TMM normalization are 10-5, In statistical standards, such p-values are typically considered statistically significant, not marginal or weak. The DiRT method obtained lower p-values, which indicates that DiRT produced a stronger statistical separation. It is recommended that this description be revised to be more objective.
- Lines 280, 323: Attributing the variation in the control group to 'age-related' in multiple places in the text seems inappropriate. This short time span during the experiment is unlikely to produce a significant age effect. It is recommended to correct this description.
- Lines 421-429: The discussion mentions that DEG-DEG pairings can also produce low p-values. The core of DiRT relies on a stable index gene; if the ratio of two regulated genes can also produce a significant signal, it will complicate the interpretation of the results. The author should explain in more detail how to systematically ensure the stability of the index genes.
- It is recommended to use methods such as RT-qPCR to validate several key genes identified by DiRT (especially genes in the new pathways).
- Line 399: Figure 5 should be Figure 6. Line 381: Table 5 should be Table 1, and the corresponding numbering in the text should also be revised.
- Line 384: Null test is not a scientific description; should it be Null hypothesis testing?
Author Response
Comment 1. Lines 240, 275: On day 3, the p-values detected for genes like Gimap6 and Mylip under RLE and TMM normalization are 10-5, In statistical standards, such p-values are typically considered statistically significant, not marginal or weak. The DiRT method obtained lower p-values, which indicates that DiRT produced a stronger statistical separation. It is recommended that this description be revised to be more objective.”
Response to Comment 1:
We appreciate the reviewer’s valuable observation. We agree that p-values in the range of 10⁻⁵ are statistically significant by conventional standards. In the manuscript, our intent was not to describe these results as statistically “weak,” but rather to note that the separation between the control and TIM samples at day 3 was relatively smaller compared with later stages (days 7 and 10), where the p-values decreased to the 10⁻¹³–10⁻¹⁸ range under the same normalization methods. To clarify this, the text has been revised as follows (lines 323-325):
“At the early stage (3 days post-tumorigenesis), both RLE- and TMM-normalized measures showed smaller differences between control and TIM samples than at later stages (7 and 10 days), although the p-values (10⁻⁵ range) remained statistically significant.”
Comment 2. Lines 280, 323: Attributing the variation in the control group to 'age-related' in multiple places in the text seems inappropriate. This short time span during the experiment is unlikely to produce a significant age effect. It is recommended to correct this description.
Response to Comment 2:
We appreciate the reviewer’s insightful comment. We agree that the term “age-related variation” may be misleading within the short experimental timeframe of this study. Our intention was to indicate differences among control samples collected at different time points (3, 7, and 10 days), rather than age-dependent biological effects. Accordingly, all instances of “age-related variation” in the manuscript have been replaced with “time point–related variation” to more accurately describe the observed differences.
In lines 329-332: “Variation associated with sampling time points was also observed in the expression of both Gimap6 and Mylip. In control samples, expression levels of these genes varied markedly across time points, showing greater variation than the differences observed between control and TIM samples at 3 days post-tumorigenesis (Figure 2).”
In lines 372-374: “Notably, whereas control measures exhibited time point–related variation under RLE normalization, DiRT normalization reduced this variability and yielded more stable values.”
Comment 3. Lines 421-429: The discussion mentions that DEG-DEG pairings can also produce low p-values. The core of DiRT relies on a stable index gene; if the ratio of two regulated genes can also produce a significant signal, it will complicate the interpretation of the results. The author should explain in more detail how to systematically ensure the stability of the index genes.
Response to Comment 3:
We appreciate the reviewer’s insightful comment regarding the potential interpretational issue arising from DEG–DEG pairings. As noted, DiRT fundamentally assumes that the index gene is not significantly affected by the target condition. In our analyses, we occasionally observed that DEG–DEG pairs with opposite regulatory directions produced DiRT values with low adjusted p-values. Such cases likely reflect artificial pairings where contrasting regulation amplifies the ratio difference, rather than biologically meaningful interactions.
To minimize their occurrence, we applied a t-test p-value threshold (> 0.05) to select index genes showing no significant response to tumor induction using adjusted p-values calculated after RLE normalization. This criterion effectively reduced the spurious pairings and ensured the suitability of selected index genes as reference candidates. Notably, such DEG–DEG pairs were rare and were previously observed only in the Drosophila DiRT dataset, where 2 of the top 10 DiRTs involved DEG–DEG combinations. No comparable instances were found in the current mouse (TIM) dataset after the filtration of index genes, indicating that this artifact has minimal influence on overall DiRT performance. We have clarified this point in the revised manuscript to explain both the rationale and the practical outcome of index gene selection.
In lines 470-484: “We observed that DEG–DEG pairs with opposite regulatory directions occasionally generated DiRT values with low adjusted p-values. Such combinations likely represent artificial pairings that coincidentally satisfied the selection criteria. To minimize their occurrence, we applied a t-test p-value threshold (> 0.05) to the index gene, using adjusted p-values calculated after RLE normalization. This phenomenon likely arises because di-viding the expression of an upregulated gene by that of a downregulated gene (or vice versa) can artificially increase the expression contrast between groups, producing exceptionally small adjusted p-values. In many practical applications, the inclusion of a small number of such pairs is unlikely to affect the main biological conclusions, as they do not dominate the ranking of top candidates. Notably, these cases were rare and were previously observed in the Drosophila DiRT dataset, where two of the top 10 DiRTs involved DEG–DEG combinations. No comparable instances were found in the current mouse (TIM) dataset after the filtration of index genes, indicating that this artifact has minimal influence on the overall performance of DiRT. Nevertheless, awareness of their potential presence remains important for accurate interpretation of DiRT results.”
Comment 4. It is recommended to use methods such as RT-qPCR to validate several key genes identified by DiRT (especially genes in the new pathways).
Response to Comment 4:
We appreciate the reviewer’s valuable suggestion to validate DiRT-identified DEGs using RT-qPCR and fully acknowledge the importance of this method for confirming gene-specific expression changes. However, the focus of the present study was to leverage the breadth and reproducibility of high-throughput RNA-seq, which has increasingly become a cost-effective alternative to RT-qPCR for transcriptome-wide analyses. RNA-seq enables simultaneous quantification of thousands of genes without the need for primer design or additional experimental resources. In our mouse tumorigenesis model, DiRT-derived expression patterns were consistently reproduced across independent time points and biological replicates, demonstrating the robustness of the identified DEGs. Furthermore, in our previous Drosophila DiRT study, cross-validation across independent RNA-seq datasets also confirmed consistent expression trends for DiRT-identified genes, supporting the reproducibility and general applicability of the method. Together, these results indicate that DiRT provides reproducible and biologically meaningful findings based on high-throughput transcriptomic data, even in the absence of RT-qPCR validation.
Comment 5. Line 399: Figure 5 should be Figure 6. Line 381: Table 5 should be Table 1, and the corresponding numbering in the text should also be revised.
Response to Comment 5:
We have corrected the figure and table numbering as suggested. Specifically, “Table 5” is now referred to as “Table 1,” and “Figure 5” has been renumbered as “Figure 6” to ensure consistency throughout the manuscript.
Comment 6. Line 384: Null test is not a scientific description; should it be Null hypothesis testing?
Response to Comment 6:
We agree that “Null test” is not the appropriate term. It has been replaced with “Null hypothesis testing” throughout the text and figure legend to more accurately describe the statistical validation procedure.
Reviewer 2 Report
Comments and Suggestions for Authors
Reviewer Comments for Shin et al
In this study, Shin and colleagues explored the use of DiRT as an alternative normalization and analysis approach to overcome the limitations of conventional normalization methods such as RLE/DESeq2 and TMM/ edgeR. This method which has been previously applied in their insect model study, represents a meaningful attempt to address the high variability typically observed in blood RNA-Seq datasets (and likely in other biological systems as well but not well discussed by authors). The authors convincingly demonstrated that DiRT improved early detection of tumor-associated transcriptional changes, and the KEGG pathway enrichment results added further biological relevance. While this methodological study could be beneficial for researchers, like me, working in this field, several important limitations need to be addressed as I have outlined below.
Major Comments
- Please provide a more detailed description of how DiRT works and how it theoretically improves upon existing normalization methods, especially considering there are multiple methods existing which they have mentioned as well. This information is essential for researchers to understand the underlying principles of this approach. Currently, the explanation in both the present manuscript and the authors’ previous publication (Insects 2025, 16(9):898; https://doi.org/10.3390/insects16090898) is a kind of limited. This lack of detail represents a major limitation. Also, I believe there are more existing methods than they mentioned, please add more description.
- Since the main claimed advantage of DiRT is its robustness in analyzing blood RNA-Seq samples, it would strengthen the manuscript if the authors could compare its performance using RNA-Seq data from other tissues or organs, particularly under complex disease conditions. In other words, does DiRT maintain better performance in more complex or heterogeneous datasets? Additional analyses will be great, or they should at least include a detailed discussion of this potential generalizability.
- Please expand the descriptions in the Materials and Methods section, especially sections 2.4 to 2.6, to ensure reproducibility and clarity.
- I wish I could see a schematic of whole experiments design.
Minor Comments
- Consider moving the subsection “3.1. DiRT normalization of TIM RNA-Seq data” to the Materials and Methods section for better structural consistency.
- The Introduction and Discussion sections lack sufficient references; please cite more related studies to strengthen the context. Give a more detailed review in this area should be helpful for the readers.
- In addition to KEGG pathway analysis, it would be valuable to include results from other functional annotation or pathway enrichment tools to provide a broader biological interpretation.
- Please discuss the potential applicability of DiRT to other datasets and experimental contexts in more detail.
Author Response
Comment 1. Please provide a more detailed description of how DiRT works and how it theoretically improves upon existing normalization methods, especially considering there are multiple methods existing which they have mentioned as well. This information is essential for researchers to understand the underlying principles of this approach. Currently, the explanation in both the present manuscript and the authors’ previous publication (Insects 2025, 16(9):898; https://doi.org/10.3390/insects16090898) is a kind of limited. This lack of detail represents a major limitation. Also, I believe there are more existing methods than they mentioned, please add more description.
Response to Comment 1:
We thank the reviewer for highlighting the need for a deeper theoretical explanation of DiRT. A new paragraph has been added to the Introduction and expanded and added new section in Materials and Methods (Section 2.6), detailing the rationale behind DiRT’s design; its stepwise computation (pairwise ratio transformation, NSD-based index selection, exclusion of affected index genes, and multiple testing correction); and how this method differs from global normalization strategies such as RLE, TMM, and ALR transformations. Additional relevant normalization literature (Evans et al., 2018; Quinn et al., 2018) has also been cited for broader context.
In lines 67-97: “Normalization methods are essential for addressing both technical and biological variability inherent in RNA-Seq data [5]. Common strategies such as counts per million (CPM), transcripts per million (TPM), and fragments per kilobase per million (FPKM) adjust for library size, sequencing depth, and gene length. More advanced statistical frameworks, including DESeq2 [6], edgeR [7], and limma/voom [8,9], further account for complex experimental designs. For instance, DESeq2 applies Relative Log Expression (RLE) normalization, which compares each sample to a pseudo-reference generated from the median gene ratios, whereas edgeR uses the Trimmed Mean of M-values (TMM) to scale libraries by estimating and trimming extreme expression values. Although these global scaling approaches are widely used, they assume that most genes are not differentially expressed across samples—an assumption that may not hold true in heterogeneous or strongly perturbed biological systems such as tumorigenesis.
Recently, we developed an alternative RNA-Seq normalization method, DEG-by-index Ratio Transformation (DiRT), which identified reproducible and biologically meaningful DEGs that were not detected by conventional global normalization approaches in Drosophila [10]. DiRT is conceptually related to compositional data analysis methods such as the additive log-ratio (ALR) transformation [11]; however, a key distinction is that ALR applies a single reference gene or global denominator to all genes, whereas DiRT performs normalization locally by forming pairwise ratios between each DEG and a condition-insensitive index gene. The index gene is selected because it exhibits similar expression dynamics to the DEG under control conditions but remains unaffected by the experimental perturbation. This design allows DiRT to preserve the true biological signal of a DEG while minimizing unwanted variation from other sources.
Unlike global normalization methods that impose a uniform scaling factor across the transcriptome, DiRT performs pairwise normalization at the gene level. This localized transformation reduces dependence on global distributional assumptions and enhances sensitivity to modest yet biologically meaningful expression differences. By first statistically excluding index genes that show significant responses to the experimental condition and then systematically evaluating all possible DEG–index combinations, DiRT identifies reproducible differential patterns that may be diluted or lost under conventional approaches”
In lines 164-183:
“2.6. Computation of DiRT Values
For each gene pair consisting of a target gene (potential DEG) and a candidate index gene, DiRT computes the ratio of their read counts (RCs). Candidate index genes are selected because their expression patterns resemble those of the target gene under control conditions but remain minimally affected by the experimental perturbation. This pairwise ratio transformation converts each gene’s expression into a relative measure referenced to its contextually similar index gene, thereby reducing sample-dependent variability while preserving the change driven by the experimental condition.
To ensure reliable index-gene selection, DiRT employs the Normalized Standard Deviation (NSD) criterion to quantify expression similarity between each target gene and its candidate index gene across control samples. Candidate index genes exhibiting low NSD values and no significant response to the experimental condition (adjusted p > 0.05, t-test with Benjamini–Hochberg correction) are regarded as suitable references. For each target gene, all valid DEG–index pairings are then evaluated, and the smallest adjusted p-value among these pairings defines the DiRT significance for that gene.
Unlike conventional global scaling methods (RLE, TMM, and ALR), which assume uniform expression stability across all genes, DiRT applies a pairwise normalization framework anchored to empirically determined, condition-insensitive index genes. This design reduces susceptibility to outlier effects, batch variation, and compositional bias, thereby improving reproducibility across independent datasets and experimental time points. Additional normalization and compositional strategies for comparison are described elsewhere [17, 18].”
Comment 2. Since the main claimed advantage of DiRT is its robustness in analyzing blood RNA-Seq samples, it would strengthen the manuscript if the authors could compare its performance using RNA-Seq data from other tissues or organs, particularly under complex disease conditions. In other words, does DiRT maintain better performance in more complex or heterogeneous datasets? Additional analyses will be great, or they should at least include a detailed discussion of this potential generalizability.
Response to Comment 2:
We focused on blood RNA-seq analysis because blood represents the most clinically accessible tissue for diagnostic applications, not because it is biologically simple. On the contrary, blood is inherently complex and heterogeneous, containing diverse cell populations whose transcriptomic composition dynamically changes under physiological and pathological conditions. Despite this complexity, DiRT demonstrated robust normalization and consistent detection of differential expression signals.
We are confident that DiRT can be effectively applied to RNA-seq data from other tissues or organs as well. In our previous Drosophila study, DiRT successfully handled transcriptomic data from whole larvae—a highly complex and multi-tissue system—showing its general applicability to diverse biological contexts. We have added a discussion on this point in the revised manuscript to clarify the potential generalizability of DiRT across tissue types and disease models.
In lines 503-511: “Although the present study focused on blood RNA-Seq data, the effectiveness of DiRT is likely not limited to this tissue type. Blood consists of diverse and dynamic cell populations, leading to substantial variability in gene expression. The ability of DiRT to consistently normalize such complex datasets indicates its potential applicability to RNA-Seq analyses from other organs or experimental systems with diverse cellular compositions. In our previous work using whole-larva RNA-Seq data from Drosophila melanogaster, DiRT also maintained stable performance, supporting its general utility in complex biological contexts. Therefore, DiRT is expected to maintain its performance in RNA-Seq data derived from other tissues or disease models.”
Comment 3. Please expand the descriptions in the Materials and Methods section, especially sections 2.4 to 2.6, to ensure reproducibility and clarity.
Response to Comment 3.:
In response to the reviewer’s request for greater clarity and reproducibility, the Materials and Methods Sections 2.4–2.7 have been thoroughly revised and expanded. The updated text now provides detailed descriptions of all analytical procedures, including the sequencing workflow on the Galaxy server, software versions, and genome references used for alignment and quantification. Specific parameter settings and filtering thresholds (CPM > 2.27), the number of evaluated gene pairs (100 000), and the exclusion criteria for index genes (adjusted p > 0.05) are now clearly stated. The revised sections also describe the statistical framework (two-tailed, equal-variance t-tests with Benjamini–Hochberg correction) applied uniformly across DiRT, DESeq2/RLE, and edgeR/TMM analyses to ensure fair comparison.
Additionally, we have included a reference to the public GitHub repository (https://github.com/shinwoongg/DiRT-normalization) containing the Python scripts used to generate the DiRT candidate database and perform statistical testing. These additions collectively enhance transparency and enable full reproducibility of the analytical workflow, allowing other researchers to independently validate and extend the presented results.
Comment 4. I wish I could see a schematic of whole experiments design.
Response to Comment 4:
We agree that a schematic diagram would help readers visualize the workflow. A new figure has been added as Figure S3 in the Supplementary Materials, providing an overview of the experimental design and the DiRT analysis pipeline.
In lines 520-521: “Figure S3: Overview of Experimental Design and DiRT Analysis Pipeline;”
Minor 1. Consider moving the subsection “3.1. DiRT normalization of TIM RNA-Seq data” to the Materials and Methods section for better structural consistency.
Response to Minor Comment 1:
We appreciate the reviewer’s suggestion to move the subsection “3.1. DiRT normalization of TIM RNA-Seq data” to the Materials and Methods Section for structural consistency. However, we believe it is more appropriate to retain this subsection in the Results Section, as it presents experiment-specific findings directly related to the TIM dataset. In the revised manuscript, we have instead expanded the Materials and Methods Section to provide a more general and detailed description of the DiRT normalization procedure while keeping the Results Section focused on the specific application and outcomes in the TIM experiment.
Minor 2. The Introduction and Discussion sections lack sufficient references; please cite more related studies to strengthen the context. Give a more detailed review in this area should be helpful for the readers.
Response to Minor Comment 2:
We appreciate the reviewer’s insightful comment. In the revised manuscript, we have enhanced both the Introduction and Discussion Sections to provide a clearer context for the study. We also added two additional references that, although not review articles, include comparative analyses of RNA-seq normalization methods. These citations help to strengthen the background and support the rationale for the development and evaluation of DiRT.
Two added references are as follows:
- Evans, C.; Hardin, J.; Stoebel, D.M. Selecting between-sample RNA-Seq normalization methods from the perspec-tive of their assumptions. Brief. Bioinform. 2018, 19(5), 776–792. https://doi.org/10.1093/bib/bbx008.
- Quinn, T.P.; Crowley, T.M.; Richardson, M.F. Benchmarking differential expression analysis tools for RNA-Seq: normalization-based vs. compositional approaches. BMC Bioinform. 2018, 19, 274. https://doi.org/10.1186/s12859-018-2261-8.
Minor 3. In addition to KEGG pathway analysis, it would be valuable to include results from other functional annotation or pathway enrichment tools to provide a broader biological interpretation.
Response to Minor Comment 3:
We agree with the reviewer’s suggestion to include results from additional functional annotation tools to broaden the biological interpretation. Accordingly, we have incorporated supplementary analyses using the Database for Annotation, Visualization, and Integrated Discovery (DAVID) [32], complementing the KEGG pathway analysis. As described in lines 494–502, DAVID integrates diverse functional annotations, including Gene Ontology terms, pathways, disease associations, and protein–protein interactions, thereby enriching the biological context of differentially expressed genes (DEGs). The comprehensive DAVID results for the 406 DiRT-only DEGs, 209 DiRT/RLE/TMM shared DEGs, and 716 RLE/TMM-only DEGs are presented in Table S2.
In lines 494-502: “We also performed additional functional analyses using the Database for Annotation, Visualization, and Integrated Discovery (DAVID) [32], complementing the KEGG pathway analysis. The DAVID Knowledgebase substantially enriches the biological context of a given gene by integrating diverse annotations, including gene/protein identifiers, func-tional domains, Gene Ontology terms, pathways, disease associations, general descrip-tions, protein–protein interactions, literature links, and small molecule interactions. The complete bioinformatic results generated by DAVID for the 406 DiRT-only DEGs, 209 DiRT/RLE/TMM shared DEGs, and 716 RLE/TMM-only DEGs are summarized in Table S2.”
Minor 4. Please discuss the potential applicability of DiRT to other datasets and experimental contexts in more detail.
Response to Minor Comment 4:
We appreciate the reviewer’s suggestion to elaborate on the broader applicability of DiRT. We have already expanded this discussion in response to Major Comment 2, emphasizing that DiRT’s robustness is not restricted to blood RNA-Seq data. Specifically, we now highlight its applicability to datasets from other tissues and experimental systems with heterogeneous cellular compositions, as well as evidence from our previous Drosophila RNA-Seq analysis demonstrating stable performance in complex biological contexts (lines 503–511).
In lines 503-511: “Although the present study focused on blood RNA-Seq data, the effectiveness of DiRT is likely not limited to this tissue type. Blood consists of diverse and dynamic cell populations, leading to substantial variability in gene expression. The ability of DiRT to consistently normalize such complex datasets indicates its potential applicability to RNA-Seq analyses from other organs or experimental systems with diverse cellular compositions. In our previous work using whole-larva RNA-Seq data from Drosophila melanogaster, DiRT also maintained stable performance, supporting its general utility in complex biological contexts. Therefore, DiRT is expected to maintain its performance in RNA-Seq data derived from other tissues or disease models.”